# Me want cookie! Towards automated and transparent data governance on the Web

Jesse Wright[1,†], Beatriz Esteves[2,†] and Rui Zhao[1,†]

[1]*Computer Science Department, University of Oxford, UK*

[2]*IDLab, Department of Electronics and Information Systems, Ghent University – imec, Ghent, Belgium*

## Abstract

This paper presents a sociotechnical vision for managing personal data, including cookies, within Web browsers. We first present our vision for a future of semi-automated data governance on the Web, using policy languages to describe data terms of use, and having browsers act on behalf of users to enact policy-based controls. Then, we present an overview of the technical research required to prove that existing policy languages express a sufficient range of concepts for describing cookie policies on the Web today. We view this work as a stepping stone towards a future of semi-automated data governance at Web-scale, which in the long term will also be used by next-generation Web technologies such as Web agents and Solid.

## Keywords

Cookie, Browser, Data Terms of Use, ODRL, DPV, Negotiation, Data Governance, P3P, Do Not Track, Reasoning, Negotiation, Solid, Web, Agents

## 1. Introduction

In the ever-evolving landscape of digital privacy, the management of personal data, including cookies, within Web browsers has become increasingly crucial. Legislative attempts to give users back control over data that is captured in cookies have largely resulted in obstructive cookie notice pop-ups across the Web containing convoluted or misleading policies, which new research suggests are not often compliant with user preferences [1]. Consequently, cookies policies and other Terms of Service agreements are deemed "biggest lie on the internet" [2].

In this paper, we present our vision of how the Open Digital Rights Language (ODRL) [3, 4], Data Terms of Use (DToU) [5] and the Data Privacy Vocabulary (DPV) [6] can be embedded into websites with RDFa, and transmitted within HTTP Headers in order to well-describe and allow negotiation over the terms of use that are applied to cookies in the browser. We argue that if deployed alongside regulatory incentives, or pressure, this technology stands to benefit individuals, industry and regulators. **Individuals** stand to benefit from a smoother experience and enhanced, semi-automated control over their privacy on the Web, with browsers managing cookie policies on their behalf. **Businesses** stand to gain significantly from this

*NeXt-generation Data Governance workshop 2024, co-located with 20th SEMANTiCS, Amsterdam, Netherlands*

[†]These authors contributed equally.

✉ jesse.wright@cs.ox.ac.uk (J. Wright); beatriz.esteves@ugent.be (B. Esteves); rui.zhao@cs.ox.ac.uk (R. Zhao)

🌐 https://www.cs.ox.ac.uk/people/jesse.wright/ (J. Wright); https://besteves4.github.io/ (B. Esteves); https://www.cs.ox.ac.uk/people/rui.zhao/ (R. Zhao)

🆔 0000-0002-5771-988X (J. Wright); 0000-0003-0259-7560 (B. Esteves)

standardised approach to cookie management. By adhering to machine-interpretable standards endorsed by regulators, companies can reduce the risk of privacy-related lawsuits, demonstrate compliance with regulations such as the General Data Protection Regulation (GDPR) [7], the ePrivacy Directive [8] or the California Consumer Privacy Act (CCPA) [9], and streamline the implementation of privacy policies. Furthermore, it becomes easier for these bodies to implement automated auditing systems that validate their compliance with their advertised terms of use. **Regulators** stand to benefit from a unified framework for describing regulations around personal data in cookies, and automated techniques for checking compliance with that regulation.

We view this work as a critical stepping stone to achieve semi-automated data governance in emerging data-centric technologies that form the next generation of the Web, such as Web Agents [10] and Solid [11, 12, 13]. In particular, browser cookies are a mature, widely used and well-understood Web technology, and the measures that websites must take to handle cookies whilst complying with jurisdictional data protection regulations are well-known. This makes browser cookies a good target use case where academia, regulators and industry can 'battle-test' and mature technologies such as ODRL, DToU and DPV to support real-world requirements for semi-automated data governance. We also note that even if cookie banners are obtrusive, cookies and their attached policies are pieces of personal data that generate privacy concerns for users. Thus we propose cookie data as the starting point for the rollout of annotated terms of use at Web scale.

The remainder of this article is structured as follows: Section 2 provides background information on Semantic Web technologies for the expression of policies and terms of use, Section 3 describes related work on privacy policies for browsers, vocabularies for expressing cookie preferences, and extensions for managing cookies, Section 4 describes our sociotechnical vision for managing personal data, including cookies, within Web browsers, Section 5 presents an overview of how those terms of use languages introduced in Section 2 can be used to express the purpose descriptions of cookies, Section 6 presents an overview of in-progress work to assess the effectiveness of those languages introduced in Section 2 for describing cookie purposes – and evaluating the effectiveness of Large Language Models (LLMs) in generating terms of use descriptions from natural language –, Section 7 presents a call to action to a range of stakeholders to collaborate on working towards the vision outlined in Section 4, and Section 8 concludes the article with a discussion on future work.

## 2. Background

This section provides background information on Semantic Web technologies for the expression of policies and terms of use, as well as how to integrate them with Web content.

### 2.1. Policy Languages

The **Open Digital Rights Language (ODRL)** [3, 4] is a World Wide Web Consortium (W3C)[1] Recommendation for the expression of policies over digital assets. It includes a standardised

---

[1]https://www.w3.org/

Information Model [3] and a Vocabulary [4] to express flexible rules over data and services. This model allows the representation of permitted, prohibited and obligatory actions over assets, which can be further limited with constraints on rules, actions, assets and parties, and duties on permissions. The vocabulary can then be used to populate different types of policies with particular actions, functional roles of parties and specific constraints, e.g., temporal or spatial. Additionally, ODRL presents an extension mechanism, through ODRL profiles, which can be used to add further terms for specific use cases. However, a few shortcomings have been pointed out in the model, mainly founded on the lack of guidance over policy enforcement [14, 15]. As such, work on a formal semantics for ODRL is under development, looking in particular at two scenarios related to access control and policy monitoring [16], with the goal of accurately describing the behaviour of an ODRL-based evaluator. Active work on the representation of ODRL policies for personal data assets is also under-way [17, 18].

Zhao and Zhao [5] proposed the concept and a realisation of *perennial* policies (denoted **Data Terms of Use, DToU**), which target the challenges and utilities of the decentralised Web, such as Solid [11]. In particular, one key goal is to support easy and smooth policy checking across applications and data providers, thus enabling users to make smooth and confident decisions on application authorisation. Building upon principles of and addressing issues of existing policy languages, the authors proposed a novel policy language containing both data's (data provider's) and application's policies; the developed reasoner supports compliance checking between them, as well as deriving policies for output data to assist continuous reasoning. They have also demonstrated how the proposed reasoning engine is integrated with Solid.

The **Data Privacy Vocabulary (DPV)** [6] is a community-based specification, being maintained and developed under the W3C umbrella, for the expression of metadata related to the processing of personal and non-personal data, based on legal requirements. DPV's main specification is a state of the art, jurisdiction-agnostic resource, containing meaningful taxonomies to describe entities, purposes, data and its processing, technical and organisational measures, legal bases, risks, rights and further privacy-related concepts. To invoke law-specific concepts, DPV 2.0 [19] currently supports concepts from EU's GDPR [7], the EU Data Governance Act (DGA) [20], the EU AI Act [21], the EU Network Information Security Directive (NIS2) [22], as well as extensions to specify personal data categories, location, risk management, technologies, justifications and AI terms. Guidance documents for the adoption and usage of DPV [23] are also available, including guides for consent records, records of processing activities, data protection impact assessments and data breach records.

## 2.2. RDFa

RDFa (Resource Description Framework in Attributes) [24] is a specification for enriching Web content with structured data, facilitating better data interoperability and search engine optimisation. It allows Web developers to embed metadata within HTML, XHTML, and XML documents using standard HTML attributes like 'about', 'property', and 'content'. By annotating elements with RDFa attributes, developers can provide additional context and meaning to the content, making it more accessible to machines, such as search engines and other data processors, which can then extract and utilize this structured information for enhanced search results, richer snippets, and improved data connectivity across the Web.

# 3. Related Work

## 3.1. Privacy Policies in Browsers

The concept of privacy policies in browsers is not new. The W3C's Platform for Privacy Preferences (P3P) [25] was a protocol published in 2002. P3P enabled Websites to describe their data management practices to users in a standardised and machine-readable format. This included a description of the type of data that was collected via different browser interactions, and the purpose for which the Website was collecting it. P3P enables Websites to encode their privacy policies in XML, which can then be automatically retrieved and interpreted by Web browsers and other user agents. This allows users to easily understand a Website's data collection and usage practices without having to read convoluted privacy policies. Moreover, it enabled the browser to block parts of the Website until users had opted into certain types of their data being collected. Despite its innovative approach, P3P faced challenges in adoption and implementation, leading to limited use and eventual obsolescence as privacy concerns and regulations evolved [26].

Global Privacy Control (GPC) [27, 28] and its predecessor Do Not Track (DNT) [29] take an all-or-nothing approach to improving user privacy on the Web, introducing a binary signal which users can enable to indicate they do not want their data to be sold or shared. Unlike DNT, GPC has gained traction because it is backed by the CCPA [9, 30]. This regulation requires companies to honour user preferences of opting out of data sharing, giving GPC more legal weight [31].

## 3.2. Vocabularies for expressing cookie preferences

Bushati et al. [32] introduced the OntoCookie ontology, developed as part of a study investigating users' awareness of the data they consent to sharing via cookies. OntoCookie is a formal representation of the cookie domain, comprising 229 axioms, 32 classes, 10 object properties, and 10 data properties. It models various types of cookies, their metadata, and their purposes (e.g., necessary, analytics, marketing) using a top-down ontology engineering approach. By leveraging this ontology, Bushati et al. [32] created a KG-based tool to enhance user comprehension of cookie data sharing, providing a more transparent and interpretable view of cookie data.

Of the 32 classes introduced by OntoCookie, 8 are subclasses of purpose for processing cookie data (Analytics, Marketing, Profiling, ServiceOptimisation, ServicePersonalisation, Service-Provision, Tracking), 10 are related to types of cookies (Authentication, HostOnly, HttpOnly, Persistent, SameSite, Secure, Session, Super, Tracking, Zombie) and 2 are subclasses of necessity, i.e., Necessary and Optional. Concerning the purpose classes introduced, only Tracking does not have an equivalent concept in DPV 2.0 (which otherwise has 88 additional purpose classes compared to OntoCookie), which may be worth adding as an extension to DPV under `dpv:Marketing`. The cookie classes are useful for adding additional descriptions about cookies, but do not help describe their terms of use, and are thus not as useful to this work.

Thus, we propose the use of ODRL, DToU and DPV in favour of OntoCookie as these vocabularies are better suited for describing terms of use in this use case, and better generalise

to terms of use descriptions outside of the context of cookie policies which is the long-term end goal of this work.

### 3.3. Deceptive patterns and extensions for managing cookies

Deceptive patterns in cookie consent popups manipulate users into agreeing to data collection. These tactics often violate GDPR principles requiring informed and voluntary consent. Common deceptive patterns include: (1) Pre-selected Options – banners with pre-ticked boxes for non-essential cookies [33], contrary to GDPR guidelines requiring active consent [34]; (2) Deceptive Button Colours – highlighting the "Accept" button more prominently than the "Reject" button to influence user choice [33]; (3) Complex Navigation – making users navigate multiple layers to be able to reject cookies, while accepting them is straightforward [35]; (4) Misleading Labels – declaring marketing cookies as essential to imply users cannot opt out without affecting functionality [35]; (5) Hindering Withdrawal – making it difficult to withdraw consent by not providing easily accessible options [36]; and (6) Manipulative Language – using vague or biased language to emphasise benefits of accepting cookies while downplaying data collection [37].

A study by Bollinger et al. [38] identified widespread GDPR violations across nearly 30,000 Websites, with 94.7% of these sites exhibiting at least one potential violation. This underscores the need for a technical infrastructure, developed in collaboration with regulators, that facilitates platforms in developing legally compliant terms of use.

To combat such problems, there have been several efforts to implement extensions which automate the process of accepting or rejecting cookies in browsers. For instance CookieBlock [38] is a browser extension designed to automate cookie consent management by using machine learning to classify cookies based one of four purposes: Strictly Necessary, Functionality, Analytics, and Advertising/Tracking. The extension then automatically accepts or rejects cookies based on which of the four categories users enable. For classification, CookieBlock achieves a mean validation accuracy of 84.4% and filters out approximately 90% of privacy-invasive cookies without significantly affecting Website functionality [38]. This approach does not depend on the cooperation of Websites, thus improving user privacy even on sites that do not comply with GDPR requirements.

As Bollinger et al. [38] acknowledge, CookieBlock, while innovative, is a temporary client-side fix. No solution limited to the client can address the root problems of (1) Websites presenting ambiguous and convoluted privacy policies that may be misinterpreted by humans and machines, and (2) Websites ignoring user consent when personal data is allowed to be used for a limited set of purposes. By nature of the CookieBlock being 'adversarial', it broke 10% of websites. Our proposal in Section 4 avoids this problem, with Websites able to describe cookie configurations required for Website functionality. Our proposal also offers more fine-grained user preferences, offering a wider range of purposes and controls for properties including the cookie retention period.

# 4. Vision

## 4.1. Overview

In our long term vision of the future Web, all data shared between clients (including browsers and other types of agents) and servers is annotated with machine-readable terms of use agreements. This enables the data sender (server or client) to declare features such as which legal basis is being used to process said data; and data subjects to express what permissions, obligations and other restrictions apply to the usage of their data. In turn, this enables the data recipient (client or server) to automatically and unambiguously determine how the data may or may not be used, using rules-based reasoning. The data recipient may also use terms of use *requests* to identify their promises and the permissions they would like the sender to include in the agreement. We expect these machine-readable terms of use *agreements* to be encoded using RDF [39], described using the ODRL [3] or DToU [5] models and to extend upon terms from DPV [6].

We now turn specifically to the case of cookie management, which is the focus of this paper. We envision migrating towards a state where the act of a data subject (in this case the user of a browser) 'permissioning' to the processing and sharing of personal data within cookies is communicated by having browsers including accompanying 'Data-Policy' header(s), containing terms of use agreements, for each HTTP Cookie header (c.f. RFC 6265) in the browser request. The contents of the terms of use agreements in the 'Data-Policy' header(s) are to be generated by the browser or a browser extension enforcing the users' data sharing preferences against the Website's request. As we shall elaborate further in the remainder of this section; this process of enforcement may either be performed statically in the browser or involve a 'negotiation' between the Website and browser agent. This process may require explicit user input depending on what existing privacy preferences the user has supplied and the legal basis used by the Website for the processing of cookie data, e.g., requiring explicit GDPR consent from users will imply their affirmative and freely given acceptance of the cookies' terms of use.

## 4.2. Pathway

We discuss a 3-step pathway towards reaching the vision, where each step itself is a valid technical solution, gradually increasing the automation and formality.

### 4.2.1. Machine readable terms of use requests in cookie dialogues

As a first step towards this goal, we propose that Websites begin embedding terms of use requests as machine-readable RDFa [24] in existing cookie dialogues. In particular, for each cookie identifying the type of data that is retained and a granular description of the purposes for which it will be processed. An example of such a request is given in Figure 1. Further, we propose the use of a standardised naming scheme for HTML **id** attributes for the checkboxes to opt in or out of particular cookies. This would enable browser extensions to automatically

manage the selection boxes on the behalf of users in a deterministic manner. This automated selection would be made based on clear preferences that the user has actively chosen. Unlike existing browser extensions which accept/reject cookies [38], this solution does not rely upon Large Language Models (LLMs) or other heuristic measures to interpret the natural language cookie policies displayed, and accept/reject cookies based on broad categories of cookies such as performance, functional and analytics. Instead, this solution allows for matching against well-defined and fine-grained user preferences in the browser.

Our proposal using RDFa embeddings would be relatively straightforward for existing Consent Management Platforms (CMPs) to independently roll-out as they are already in control of the current pop-up functionality. Having a browser extension interact with the existing HTML element, rather than posting data directly to an API also means that there is no interim effort required to align the consent management APIs that CMPs use.

Similarly to Bollinger et al. [38], limited client (browser) side enforcement of user preferences may be introduced once embedded RDFa descriptions are available. In particular, the browser may prevent the creation of cookies which may not be used for any purposes according to user preference. At this stage of the work this simply means that the only cookies permitted on a given Website are those that have been 'allowed' by the extension.

For those websites that do not move to use RDFa embeddings to describe the purposes; there is also the option of using LLMs called by a browser extension in the short-term to translate the natural language privacy policies into machine-readable terms of use requests. The experiments we propose in Section 6 will determine the viability of this approach. These formal descriptions can then be used by browser extensions to automatically accept or reject cookies based on user preferences. This LLM-based supplement for RDFa embeddings is an ad hoc solution in that it does not address the root problem of ambiguous and complex natural language privacy policies, but rather provides a temporary fix to the problem of manual cookie management. However, it would improve the granularity of control that users have in comparison to existing solutions such as CookieBlock [38].

### 4.2.2. Machine-readable terms of use requests in headers

Over time, we recommend that Websites also include these terms of use requests as HTTP headers with the name 'Data-Policy-Request', which contains either the full privacy policy for Website cookies, links to a cacheable description of the privacy policy, or links to an API for policy negotiation (c.f. Section 4.2.3). Whenever both a browser and Website implement the 'Data-Policy' detailed in Section 4.2.3, there would no longer be a need for server-managed cookie user interfaces.

Unlike Section 4.2.1, even Websites using CMPs will need to take individual action to implement this header-based proposal, as CMPs do not generally intercept response headers. Consequently, adoption is likely to be slower.

### 4.2.3. Consent and machine-readable terms of use agreements

In parallel to the recommendations of Sections 4.2.1 and 4.2.2, we suggest browsers start implementing the 'Data-Policy' header mentioned in Section 4.1. This allows browsers to

communicate to Websites, on the behalf of users, the permissions that users have given for the processing and sharing of their personal data collected in cookies. This role is currently fulfilled by Consent Management Platforms (CMPs), which use custom APIs and flows to collect and log user consent.

This proposal is backwards compatible with HTTP servers and browser clients that do not recognise 'Data-Policy' headers as "a proxy MUST forward unrecognised header fields" and "other recipients SHOULD ignore unrecognised header and trailer fields." [40]. We propose the name 'Data-Policy' rather than 'Cookie-Policy' in anticipation that the header will also transmit terms of use agreements for other non-cookie headers and the message body in the future.

With 'Data-Policy' headers, browser clients can enforce terms of use agreements which do not permit the cookie data to be used for any purpose, by blocking the cookie from being sent. Clients will not be able to enforce terms of use agreements that allow the Website owner or third parties to use personal data for a limited range of purposes. In these cases the Website and third parties are responsible for adhering to the agreed terms of use; similarly to how Websites and third parties are expected to respect consent signals sent out by CMPs today. Rather than relying on platforms benevolently respecting these terms of use agreements, our goal is to work with regulatory authorities to make these terms of use agreements legally enforceable. In EU countries and the UK, we can start by making 'Data-Policy' headers the recommended way to signal legal consent to process personal data. In the future, we envision these terms of use agreements being of a more contractual nature, similar to a data sharing agreement. As we shall re-iterate in the following subsections and Section 7, this is where a joint approach is required between research, regulatory bodies and industry, in order to concurrently develop technical standards, new regulations and best practices to:

1. Ensure that with this proposal, companies can comply with EU and UK regulation for consent collection and logging [41] which is currently fulfilled by custom flows that take place when users confirm their preferences in cookie consent dialogues.

2. Provide regulatory incentives or pressures for adoption of the 'Data-Policy' header. Incentives could include "Safe Harbour"[2] [42] clauses in data protection regulation, which legally protect companies using policy evaluation engines to ensure terms of use agreements are respected. As a more stringent measure, regulations could mandate the use of such policy engines and require companies to undergo regular audits to verify their compliance and obtain certifications from Data Protection Authorities.

3. Align this proposal with existing semi-automation approaches adopted for data governance within enterprises [43].

4. Ensure this proposal enables the reduction of long-term engineering, legal and other compliance-related costs that enterprises face [44].

### 4.2.4. Negotiating terms of use agreements

We anticipate use cases where terms of use agreements cannot be immediately computed by browsers using static terms of use requests from Websites and user preferences (that can be

---

[2]A safe harbour provision is a legal clause that offers protection from liability or penalties under specific conditions, as long as the party complies with certain predefined guidelines or standards.

either pre-defined or obtained from the user in real-time via a browser-managed popup). For instance, some Websites offer a choice between paying for services and consenting to the use of personal data for targeted advertising [45]; others may present ads at a lower frequency when targeted advertising is possible [46]. Such Websites may need to offer a 'negotiation' API where browsers can perform operations such as payments and obtain a service contract, complementing their terms of use agreement, that allows use of the Website without sharing data for targeted advertising.

As discussed by Solove [47] and Florea and Esteves [48] obtaining valid GDPR consent remains an issue, as it implies the user knows the purpose for which their data is being used, as well as the identity of the legal entity 'behind' the processing of said data, among other conditions. As such, the development of a policy-based Web environment must not shy away from going beyond consent to explore other legal bases, while, of course, relying on it as an information safeguard for users.

### 4.2.5. Beyond Cookies

We are striving for a future in which all data sent over the Web is annotated with terms of use agreements between the sender and recipient. This could be achieved by having HTTP requests and responses always containing the 'Data-Policy' header. This header would contain terms of use agreements that would be applicable for data in the headers and message body with the possibility to have fine-grained definitions of different terms of use for different parts of the request or response. This enables the sender to be explicit about any requirements they have for how their data is governed, and for the recipient to implement policy engines that ensure these requirements are respected. In the worst case, if a server is not able to understand or handle the terms of use of an incoming HTTP request, we would expect the server to return a 5xx response and discard of any user data received in the request. It would be best practice for a server to also indicate any modifications required to the terms of use in order for a future request to be accepted, using an RDF-encoded response. If a client receives a set of terms of use that it is not able to respect, it should also immediately discard of the information received in this response.

### 4.2.6. Beyond Websites

In the spirit of the Semantic Web [10, 49, 50], we posit that over time the Web will evolve away from users sending and receiving data via Websites, to having their interactions on the Web mediated via Web agents such as Charlie, the "AI that works for you." [51]; with most user information stored across personal data stores such as Solid Pods [11, 12, 13].

In this vision, we hypothesise that all data sent from personal data stores, and between personal agents will need to be annotated with terms of use to enable automated compliance with data governance requirements; as data is sent between agents representing data subjects with a range of preferences and legal rights, and hosted in personal data stores across a range of legal jurisdictions. These, and a range of other factors, will influence the terms of use that recipients must comply with when receiving data they wish to process. Specifically, unlike cookies on websites, secondary data transmission (of original and downstream data) and data

combination are expected to be more prominent, both in spatial and temporal manners, in this interaction model, which requires special attention in policy languages and engines.

Our hope is that by introducing a 'Data-Policy' header where agreed-upon terms of use can be exchanged between clients and browsers; we prove the concept of terms of use agreements at Web scale; and this 'Data-Policy' header can be extended and re-used to support HTTP-based data sharing between Web agents, personal data stores and data processors.

### 4.3. Benefits to different stakeholders

#### 4.3.1. Benefits to users

The proposed solution promises to enhance user experience and privacy.

On the Web today, users are forced to choose between (1) reading through and interpreting extensive lists of cookie policies in order to manually accepting or rejecting the cookies a Website has, (2) accepting or rejecting blanket lists of cookies such as functional, performance and analytical; with the UI to do so often only available after navigating deceptive UX patterns [35], or (3) give in to accepting all cookies in order to move to using the Website as quickly as possible. In all three cases, a user's experience of the Web is interrupted, resulting in a disrupted user experience.

With users able to pre-set their privacy preferences, or have them remembered by the browser across Websites, the obstructive nature of cookie popups is reduced. This also ensures that users' privacy preferences are consistently applied without additional effort, thereby enhancing their control over personal data.

By having terms of use requests and agreements encoded in a machine-readable format, there is less ambiguity over what purposes users are permitting their data to be used for. Having the recording of consent managed directly by the browser reduces the opportunity for Websites to introduce deceptive UX patterns.

#### 4.3.2. Benefits to implementors

The implementation of standardised cookie management policies is anticipated to provide several benefits to businesses.

First, a "safe harbour" [52] provision could be established for companies that adhere to these standards, potentially reducing their risk of facing regulatory fines or lawsuits [53]. In particular, regulators or Data Protection Authorities could offer automated compliance checks to aid Websites in verifying that the machine-readable terms of use requests they make, and the terms of use agreements they accept, are consistent with jurisdictional regulations such as GDPR and CCPA. By tagging all information incoming to their system with the applicable ODRL or DToU policies, businesses can use policy evaluation engines to confirm that they are using personal data within the scope of permission, prohibition and obligations that have been granted. A safe harbour clause may be available in cases where companies are able to produce internal audits demonstrating faithful use of these compliance-checking tools.

Secondly, the ease of implementation is a significant advantage. Developers would find it simpler to generate privacy policies based on their system architecture, allowing for more accurate descriptions of data usage purposes. This streamlined process not only facilitates

better compliance but also reduces costs. Legal teams would only need to review the selected policies, rather than drafting comprehensive terms-of-service documents from scratch. When policy changes (e.g., introducing a new functionality), the legal team can easily understand the difference from the comparison between old and new formal policies. In addition, they may have internal compliance tools built around the automated reasoning of such formal policies. This efficiency can lead to significant savings in both time and resources for businesses.

Furthermore, there are flow-on effects of the benefits provided to users. For instance, platforms are likely to have higher retention rates if users are able to receive tailored online experiences without being concerned that their information is being used for purposes they are not comfortable with.

### 4.3.3. Benefits to regulators

Our proposal offers a number of benefits to regulators. With formal descriptions of cookie policies, there is a possibility of verifying whether Website policies are coherent with local regulations in a semi-automated manner. This would likely reduce the cost of having legal experts do this process manually. Furthermore, the use of machine-readable terms of use agreements would improve the accuracy and efficiency with which regulators ensure that companies adhere to the agreements they make with users. This is because companies would be able to present system audits with standardised reports, which would be easier for regulators to review.

### 4.4. Comparison to Consent Management Platforms

Today, most Websites use Consent Management Platforms (CMPs) to manage their data privacy obligations related to cookies. Consent Management Platforms (CMPs) have 4 primary roles [54]: **Consent collection** via cookie banners; **Consent management** blocking cookies which users have not consented to the use of; **Consent signals** sharing the collected consent with first- and third-party data processors, such as analytics platforms and ad vendors; **Proof of consent** storing proof of consent for regulatory purposes.

For the sharing of consent signals, each third party vendor implements custom consent API's such as the Google Consent API, and CMPs bear the responsibility of translating the collected user consent into a fixed set of consent options offered by the 3rd party vendor. Our proposed introduction of the 'Data-Purpose' header offers a better experience for both vendors and end users. Browsers become responsible for **consent collection** and **consent management** of cookies; **consent signals** are sent directly from the browser to first and third parties via 'Data-Purpose' header, removing the need for intermediary management of consent signals; and **proof of consent** is made possible by simply maintaining a log of the 'Data-Purpose' headers that Websites receive. Consequently, our proposal makes the flow of user consent records more rigorous, both by having unambiguous machine-interpretable records of the terms of use that users have agreed to; and having these agreements sent directly to the relevant data processor rather than requiring out-of-band communication via CMPs.

## 4.5. Comparison to related work on privacy policies in browsers

The core differences between our proposal and the related works have to do with the expressivity of the languages that we propose to use, and the differing legal context at the time in which we do our work.

**P3P** [25] contains many similar concepts to those which we propose here, including having the ability to define cookie purposes (although from a fixed vocabulary) and a trust engine to mediate between user preferences descriptions of cookie purposes. Regarding expressivity, P3P proposes describing the following features of cookies: (1) Categories – what information is collected, (2) Purpose – how it is used, (3) Recipient – who has access to it, (4) Retention – how long it is stored, and (5) Access – what information can the user access. ODRL and DToU presented in Section 5 express a superset of these concepts. P3P only offers 10 purpose categories (and one 'other') category that are built into the specification and thus not extensible. In contrast, DPV currently has 95 purposes, and is extensible through OWL [55] – with the ability to preserve semantic relationships. For instance, Websites could define 'marketingDigitalProducts' as a subclass of 'marketing' to improve precision of their terms of use requests; and browsers would still be able to apply all user preferences for 'marketing' preferences to the request.

We recognise that there have been attempts to extend P3P with policies modelled in RDF [56] using Rei [57]. The primary goal of this work by Kolari et al. [56] was to improve expressivity and adoption of P3P. The primary advantage of our proposed use of ODRL or DToU with DPV is (1) ODRL and DPV are more mature than Rei [57], (2) DPV has an extensive range of terms available for describing a wide array of privacy concepts, (3) DToU provides a unified framework covering more concept scopes envisioned in this paper and (4) these vocabularies are built with modern regulation, such as GDPR [7], in mind.

Compared to P3P, our proposal changes the party controlling the terms of use agreement. With P3P, Websites declare the privacy policies associated with different components of the site; similar to the terms of use requests that Websites make in our proposal. However, in P3P the browser is not able to respond with a modified agreement when it sends data, which may choose to allow the Website to use the data for a subset of the purposes it had requested. Instead, in P3P the browser only has the option to block the widget which sends the data; making that functionality completely unavailable to the user.

The **Do Not Track** (DNT) and **Global Privacy Control** (GPC) initiatives are less expressive by design, only offering a single signal to indicate that users do not wish to be tracked via cookies.

In terms of the legal context, there is some level of consensus that P3P and DNT were both unsuccessful due to a lack of legal pressures, however, as evidenced by the greater success of Global Privacy Control, there is a greater promise of adoption for such technologies when they are mandated in regulation such as the CCPA. This is why we do not propose an isolated technical solution, but rather a collaborative development between research, industry and regulatory bodies to work towards a sociotechnical solution by which the vocabularies used for formally describing terms of use agreements between the user and the website contain terms and concepts that have a well-understood legal interpretation.

## 5. Describing cookies using ODRL and DToU

We now discuss how cookie policies can be described using ODRL and DToU, respectively. In this paper, we do not aim to prematurely conclude which of these vocabularies is best suited to describe cookie purposes and terms of use agreements between users and Websites. Instead, we provide an overview of the strengths and weaknesses of the two languages for modelling cookie policies. In later sections, we propose future work which we expect will shed light on which language is most suitable for which use case, and what modifications, if any, are required for the languages to be adopted. In particular, we expect to be informed by (1) performing experiments such as those outlined in Section 6, to test how well natural language cookie policies can be encoded in these formal languages; and by (2) co-designing with regulators and industry as discussed in Section 7. As such, we shall present now how a cookie policy with the purpose description "Download certain Google Tools and save certain preferences, for example the number of search results per page or activation of the SafeSearch Filter. Adjusts the ads that appear in Google Search" is expressed using ODRL and DToU.

When it comes to the representation of cookie information as ODRL-based policies, such as that shown in Figure 1, two advantages can immediately be described in terms of flexibility and extensibility. In this context, ODRL has the flexibility to model distinct concepts embedded in the cookie description in human-readable language as machine-actionable elements, namely in terms of actions (processing operations) and purposes – this flexibility was already demonstrated for the particular use case of health data sharing by Pandit and Esteves [18]. Moreover, for the inclusion of new terms, ODRL provides extensibility through its profile mechanism – as shown by the usage of the ODRL profile for Access Control (OAC) which allows the expression of legally-aligned policies with DPV [17]. As for shortcomings, it should be pointed out that, by design, ODRL does not allow the expression of dynamic constraints, although a solution using property paths is being proposed to deal with this issue [15]. The resolution of this limitation is of particular importance for the modelling of temporal constraints and for cookies specifically when their retention period needs to be enforced or in case it needs to be updated.

On the other hand, DToU modelling features a standard mechanism to represent both the cookie policy (as an app policy), such as that shown in Figure 2, and the user's preferences (as a *data* policy), as well as the formal semantics behind the modelling language. Thus their compliance can be verified directly using Notation3 reasoning [58, 59, 60]. It provides an extensible *tag* mechanism (whose extension functions similar to profiles, though informally), which can be used to model the *purpose* constraints, as demonstrated in Zhao and Zhao [5]. Through enumeration and subclassing, the tag mechanism can be used to represent (discrete) temporal constraints – for example, by having longer durations as super-classes, and shorter durations as sub-classes of them, temporal constraints can be modelled as a *requirement*, one of the two core types of tags. However, a proper mechanism may be needed to represent temporal information without enumeration, particularly through extending the existing prohibition and activation condition mechanisms. Furthermore, some informational fields are missing from the current DToU policy language, such as creator and creation time.

# 6. Analysing the effectiveness of ODRL and DToU with DPV for describing cookie policies

In this section, we propose a methodology that we are currently implementing in order to analyse the effectiveness of ODRL and DToU with DPV for expressing platform cookie policies as terms of use requests. Our progress towards implementing this methodology is available at https://github.com/jeswr/cookie-analysis/tree/chore/noise.

## 6.1. Dataset

For our analysis, we use a dataset of approximately 304,000 cookies which Bollinger et al. [38] collected to perform their work on CookieBlock. These cookies were collected from around 30,000 websites that use Consent Management Platforms (CMPs).

Bollinger et al. selected CMPs that list cookies with their purposes. They then extracted cookies declared by the CMPs and those created during interactions with the Websites. This process resulted in a comprehensive dataset of declared and observed cookies. The dataset includes around 304,000 cookies, with details such as their names, domains, expiration times, purposes and categories.

## 6.2. Research Challenge

For most properties, we are able to define a static rules-based mapping from this schema into ODRL and DToU descriptions similar to those shown in Figure 1 and Figure 2. The key property which requires linguistic interpretation is the *Purpose.* In particular, the used dataset contains a natural language description of the purpose for which a cookie is being collected, whilst in ODRL and DToU we seek to map this to a list of well-defined DPV purposes to associate with the `odrl:constraint` and `dtou:purpose` properties, respectively. Consequently, we seek to experimentally answer the research questions:

1. Does the current DPV Purpose vocabulary contain all concepts required to describe cookie purposes, if not, what new concepts are required?
2. Is it possible to accurately automate the mapping from natural language descriptions of purposes to DPV descriptions? This would allows us to establish (a) the viability of having browser extensions generate terms of use requests in the short term, and (b) the extent to which Website owners can automate the generation of terms of use requests from their existing privacy policies.

In the following sections, we describe an experimental methodology which we are in the process of implementing in order to answer these research questions.

## 6.3. Methodology

We use a SPARQL [61] query engine to dereference the DPV ontology and query for the definition, label and note of all `dpv:Purposes` according to the SPARQL query found here.

From this, we generate a document listing all of the definitions, labels and notes with an anonymous ID. This document can be found here. For each cookie we wish to classify, we

pass this document to an LLM along with the name, category and description of the cookie. We prompt the LLM to identify the IDs of any relevant DPV purposes, if there is a part of the cookie description that is not captured by the current purposes, we ask the LLM to propose a description of a new DPV purpose that it would use in the description of the cookie purpose. The prompt used for this generation is available here. A sample response from the LLM can be found here. Note the explanation is requested to encourage the LLM to perform chain-of-thought reasoning when performing purpose classification; and this explanation is not used elsewhere.

We are in the process of analysing the LLM proposed purposes to develop a new dataset of purposes to be proposed to DPV. The methodology we are planning to apply to achieve this is as follows:

1. For all of the new DPV purposes proposed from analysing the 304,000 available cookies we insert the purpose description into a vector database [62].
2. Group purpose descriptions by embedding similarity.
3. For each group of purposes with high similarity, pass that set of descriptions back to the LLM and prompt it to (a) propose a purpose description that aligns with the input set, (b) propose a name for the new purpose, (c) identify whether the proposed purpose is a subclass or superclass of any of the existing purpose descriptions, and (d) output this information according to a template `.ttl` document.

## 6.4. Proposed Evaluation

We propose that the quality of the LLM classification and generation of purposes be assessed by a set of legal experts. For this evaluation, a random sampling of the cookie dataset will be selected and legal experts will be provided with the cookie purpose descriptions, and asked to (1) select the set of DPV purposes they believe apply, and (2) describe any concepts they believe are missing.

If both they and the LLM have identified that there are no DPV concepts to accurately describe the cookie purpose, the legal expert will then be asked to identify whether the new concept(s) proposed by the LLM match the concept(s) they propose.

# 7. Call to action

## 7.1. Regulators

We call upon legal and policy experts working in the space of data governance to participate in co-designing machine-readable cookie description and transmission standards. In particular, we call upon supervisory bodies, such as the European Data Protection Board (EDPB) or the Information Commissioner's Office (ICO), that are capable of enforcing compliance with standards such as those we propose for terms of use descriptions, to ensure compatibility between the architectures we build, and the regulatory frameworks of the regions in which they will be deployed.

In the European context, we would like to work with EDPB to understand how these transmitted terms of use can become legally binding Data Sharing Agreements (DSAs), or be considered lawful consent for data processing by data subjects. The ideal outcome of this work-item is

a set of terms, and flows, which all EU Data Protection Authorities recognise as valid DSAs or lawful consent under GDPR [7]. In the UK, we would like to work with the ICO to work towards a similar understanding of how transmitted terms of use can constitute legally binding DSAs or lawful consent under the UK Data Protection Act 2018 (c. 12) [63].

A concrete starting point is to establish how terms of use annotations in the 'Data-Policy' header sent by browsers can constitute lawful consent for data processing. In particular asking the question: how do we ensure that these terms of use annotations constitute lawful consent when a browser or browser extension computes these terms of use annotations on a users' behalf?

### 7.2. Industry

At the same time, we call upon industry, including CMP providers, to co-design a solution that will benefit the customer experience and also add value to companies that implement the standard. Secondarily, we call upon industry and research centres, such as the Joint Research Centres (JRCs) supported by the European Commission, that have experience implementing mechanised data governance to participate in implementing and validating the proposed policy languages for terms of use exchanges. In particular, we seek to reduce the friction in the adoption of this Web standard by having the formal policy descriptions easily map to internal architectures for enterprise data governance.

## 8. Conclusion and Future Work

In this paper, we presented a comprehensive vision for the future of automated and transparent data governance on the Web, specifically focusing on the management of cookies. Our proposed solution leverages policy languages such as ODRL, DToU, and DPV to describe cookie policies and data usage agreements in a machine-readable format. This approach aims to enhance user control over their privacy, streamline compliance for businesses, and facilitate regulatory oversight.

The immediate next steps involve completing the evaluation outlined in Section 6 to validate the capability of existing technologies in expressing cookie policies. This will involve detailed testing and refinement of our methodology to ensure robustness and accuracy.

In parallel, we plan to engage with legal, policy, and industry experts, as discussed in Section 7, to test the real-world viability of our proposed solution. Collaboration with regulatory bodies such as the European Data Protection Board and the Information Commissioner's Office will be crucial in ensuring that our framework aligns with regulatory requirements and can be effectively enforced.

Ultimately, such collaboration and reformation will establish a standardised semi-automated approach to data governance at Web scale that benefits all stakeholders – users, businesses, and regulators – and paves the way for more transparent and efficient management of personal data on the Web.

## Acknowledgements

Jesse Wright is funded by the Department of Computer Science, University of Oxford. Beatriz Esteves is funded by SolidLab Vlaanderen (Flemish Government, EWI and RRF project VV023/10). Rui Zhao is funded by the Ethical Web and Data Architecture in the Age of AI (EWADA) project, whose funds come from Oxford Martin School, University of Oxford.

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

# 9. Appendix

```
@prefix odrl: <http://www.w3.org/ns/odrl/2/>
@prefix dcterms: <http://purl.org/dc/terms/>
@prefix dpv: <https://w3id.org/dpv#>
@prefix oac: <https://w3id.org/oac#>
@prefix ex: <http://example.com>

<https://example.com/cookie-policy-grooveshark> a odrl:Request ;
   odrl:uid "8dc5d7e3-e31f-421a-8bad-6540172d787f" ;
   dcterms:description "Download certain Google Tools and save certain preferences, for
       example the number of search results per page or activation of the SafeSearch
       Filter. Adjusts the ads that appear in Google Search." ;
   dcterms:creator ex:google ;
   dcterms:issued "2024-06-03T17:58:31"^^xsd:dateTime ;
   odrl:profile oac: ;
   odrl:permission [
       odrl:assignee ex:google ;
       odrl:action oac:Download, oac:Store, oac:Profiling ;
       odrl:target <https://example.com/grooveshark-cookie-data> ;
       odrl:constraint [
           dcterms:title "Purpose for processing is to conduct marketing in relation to
                   organisation or products or services." ;
           odrl:leftOperand oac:Purpose ;
           odrl:operator odrl:isA ;
           odrl:rightOperand dpv:Marketing ] ;
       odrl:constraint [
           dcterms:title "Rule can be exercised in the next 2 years." ;
           odrl:leftOperand odrl:elapsedTime ;
           odrl:operator odrl:eq ;
           odrl:rightOperand "P2Y"^^xsd:duration ]
   ] .
ex:google a dpv:DataController ;
   dpv:hasName "Google" ;
   foaf:page <google.com> .
```

**Figure 1:** A hand crafted ODRL description of the cookie policy

```
@prefix dtou: <urn:dtou:core#>.
@prefix dpv: <https://w3id.org/dpv#>.
@prefix dur: <http://example.com/duration#>.
@prefix ex: <http://example.org/>.

ex:ap a dtou:AppPolicy;
    dtou:name <https://url-to.website/>;
    dtou:input_spec ex:cookie1 .
ex:cookie1 a dtou:InputSpec;
    dtou:data <https://example.com/grooveshark-cookie-data>;
    dtou:port [ dtou:name "google-cookies" ];
    dtou:purpose [ dtou:descriptor dpv:Marketing ];
    dtou:expect [ dtou:descriptor dpv:Download ],
        [ dtou:descriptor dpv:Store ],
        [ dtou:descriptor dpv:Profiling ];
    dtou:provide [ dtou:descriptor dur:two-year ];
    dtou:downstream [ dtou:app_name <https://google.com>; dtou:purpose dpv:Marketing ].
```

**Figure 2:** A hand crafted DToU description of the cookie policy

