# OpenReview forum: "Me want cookie! Towards automated and transparent data governance on the Web"
_SEMANTiCS.cc/2024/Workshop/NXDG — NXDG 2024_

### Official Review · ~Victor_Morel1 · 2024-07-18
**The paper may presents a relevant topic for discussion on a timely issue, but falls short in terms of clarity, presentation, and rigour, thus preventing an efficient communication of its ideas to the reader. However, a careful overhaul of the document can result in a decent contribution.**

**Rating:** 6
**Confidence:** 4

**Review:**

### Summary
This paper presents a "vision" to automate personal data management on the web using policy languages in web browsers.
The approach is exemplified with two languages, and the sketch of an evaluation is presented.

### Overview
The paper may presents a relevant topic for discussion on a timely issue, but falls short in terms of clarity, presentation, and rigour, thus preventing an efficient communication of its ideas to the reader.
However, a careful overhaul of the document can result in a decent contribution.

### Philosophy of the paper
On a general note, the vision is not very convincing, it remains quite abstract, and especially Section 4, which lacks a proper structure (3 pages before the first subsection).
Perhaps the addition of an overview followed by a more detailed presentation could help.
Also, the envisioned negotiation sounds sketchy at best): is it intended to simply face users with a choice to either pay or consent? This approach already exists, with consent rates above 99% (see https://arxiv.org/abs/2309.11625).

What is the feasibility of the approach, does the approach implies adding all this metadata in HTTP headers?
What would be such an overhead? Could it be implemented as a browser extension for instance?

Also, which actionable steps does the paper foresee to have its vision adopted by the different stakeholders mentioned (users, controllers, regulators)?
The call for action sounds like wishful thinking, more concrete action is required.

The paper mentions "going beyond consent", I advise to read Murky Consent by Solove (https://papers.ssrn.com/sol3/papers.cfm?abstract_id=4333743) about the limits of existing consent models.

### Per section
The introduction is not informative, it reads "we embed X in Y with Z, and voilà": it needs to introduce the key concepts.

The distinction between the Background and the Related Work is not clear: why is there DPV in the Background but a dedicated section to vocabularies in the Related Work? Also, because the bulk of the paper hasn't been presented, I recommend putting a real comparison of the Related Work after the introduction of the vision (instead of spreading bits of related work in Section 2, 3, and 4).

Be systematic when claiming benefits (see Section 4.2 for instance), it's hard to see the link the connection between the limitations and the benefits.

Why is Section 5 a section in itself? It is short and little descriptive, it might better fit a subsection of the vision.
Also, is Section 7 implementing this "bandaid" approach? If yes please specify it.

Section 6 claims an assessment of strengths/weaknesses of the two solutions but the comparison is again hard to see, please provide a clear summary.

I don't understand the evaluation suggested in Section 7, what are you evaluating, whether DPV has enough purposes?

## claims
Please find here a list of unsupported (or wrongful) claims:
- What does the paper mean by "In particular, this is a problem space which can be immediately addressed with collaboration between academia, regulators and industry"?
- What does the paper mean on page 4 by GPC not being "yet" enforced by GDPR?
- The paper envisions repeteadly to automate consent, this is arguably illegal under the GDPR (see https://arxiv.org/abs/2305.08747)
- What does the paper mean by "once embedded RDFa descriptions are available"
- The paper criticizes CookieBlock, I suggest to read their latest research for a more nuanced critique https://doi.org/10.56553/popets-2024-0012

## typo/language
While not necessarily impairing the reading, a few typos and language issues (sometimes undefined terms) are listed below:
- please number the pages if possible
- consent pop-ups, terms of use, terms-of-use policies, terms-of-use agreements: please make a choice amongst these terms
- automated, semi-automated, (semi-)automated? Same as above: pick one term, define it, and stick to it
- don't use "the authors", or at least consistently with "we"
- "containing long-winded policies which often are often" duplicate of "often"
- page 2, "the" GDPR (which is only spelled out on page 3 BTW)
- "(a.k.a. Data Terms of Use, DToU)" don't use "a.k.a" in academic writing, prefer "denoted" or "called"
- dark patterns --> deceptive patterns (see https://www.deceptive.design/about-us)
- ."Global Privacy Control [27, 28] and its’ “Do Not Track” (DNT) [29] predecessor" --> "GPC and its predecessor DNT"
- don't use "you" in academic writing (page 7, "to use the Website you opt in"), prefer "the user" or any other impersonal form
- page 8, what is "private" data?
- page 9, what are "implementors"? and a "“safe haven” provision"

---

### Official Review · ~Anelia_Kurteva1 · 2024-07-22
**Improtant and interesting topic for multiple communities. A good discussion on the current status of cookies and data collection on the web. However, the paper content itself can be more concisely and clearly presented.**

**Rating:** 6
**Confidence:** 4

**Review:**

This was an interesting paper on a timely topic, which  I enjoyed reading.  The need for such semantic-based approach to cookie policies is well-justified. I think the approach fits the scope of the workshop and will raise an interesting discussion. For the paper itself, I have the following comments that if resolved will significantly improve it.

Major comments:
In my opinion, the paper would be better (in terms of content presentation) if sections 2 and 3 are combined. One can start with an overview of privacy policies in web browsers and then mention the technologies that currently exist, their limitations etc.

In section 3.2, an older paper that can be interesting to the authors is https://www.w3.org/2001/sw/Europe/events/foaf-galway/papers/fp/semantic_soup/.

Since the authors discuss dark patterns, some specific software solutions for cookie banners can be mentioned as well. See OneTrust (https://www.onetrust.com/products/cookie-consent/) and CMP (https://www.consentmanager.net/cookie-banner/).

Where are the statistics for CookieBlock in section 3.3 taken from? Add references if this analysis was not conducted by you.

Towards the end of section 3, the authors mention "our proposed solution", however, it is not exactly clear to the reader what this solution is. The authors can better highlight their vision and contribution in section 3 and make a connection to section 4.

In section 4, at the beginning, the sentence about the vision is not completely clear. Better to paraphrase the sentence and make it simple in structure. Even now most website present their privacy policies, however, their comprehension and accessibility are questionable.

In section 4, who are the clients? I can envision that different clients have different knowledge of web privacy and this is a challenge (as it is with informed consent) when asking them to review terms and conditions. I am curious to know who the authors think should implement the metadata within the browsers and websites. Would this be the front-end developer working for a company or this might be a plug-in that end-users use with their browser? How can we ensure the validity and trustworthiness of the metadata?

It would be useful to showcase your cookie approach in section 4 with a graph (e.g. a UML).

To the benefits of using your approach and adding semantics, one can also add "discoverability" as a benefit. See https://link.springer.com/chapter/10.1007/978-3-319-28231-2_1.

The paper can be a bit more concise, as now it took 5 sections to present the actual proposed approach.

Sections 2.1 and 6.1 have the same title.

Would the authors envision that the decision to accept/reject a cookie is added to the cookie policy or kept separately? How would deleting browser cookies and having sessions affect the collection of data if cookie policies are implemented as suggested?

The start of the paper mentions that the solution is not LLM-dependant, however, it seems as it is based on the proposed methodology. What can be used instead of LLMs?

I can see the benefit the authors' solution will bring, however, I would have expected more concrete discussion on what happens after the solution is implemented - aka the management of the cookies. What might be the challenges?

In section 9, the authors mention "e presented a comprehensive vision for the future of automated and transparent data governance on the Web, specifically focusing on the management of cookies". I believe only one part of management - the creation and semantic specification of cookie policies is discussed. Management comprises many more stages or processes. For example, who will be responsible for managing all browser cookie policies following this approach?

I am missing the connection to SOLID and decentralisation towards the end of the paper. I would suggest taking a step back and think how to better present the idea. For example, showcase ideas of implementation in both centralised and decentralised use cases and make comparison between both.

The conclusions (section 9) do not discuss the envisioned or expected limitations of the proposed approach.

Minor comments:

The last paragraph in section 1 is too long structure-wise. Too many ",".
ODRL should be spelled out when 1st mentioned.
Section 2 should be introduced with at least 1 sentence before presenting the subsections.
Missing reference to ODRL at the start of section 2.1.
Missing reference after Zhao and Zhao at the start of section 2.2.
In section 2.2. use "the authors" rather than "they".
DPV should be spelled out when 1st mentioned.
Missing reference/footnote to DPV 2.0 in section 2.3.
GDPR should be abbreviated when 1st mentioned.
Missing references to the mentioned laws in section 2.3.
W3C should be spelt out at the start of the paper and the term should be consistently used after that (see my previous comments on this).
In section 3.1, quotes should be in italics and referenced accordingly.
Not clear why there are quotes around Do Not Track in section 3.1.
Missing reference to GPC.
Repetition of Do Not Track - use the abbreviation instead in section 3.1.
No need to cite the full title of [33]. One can add the ontology Github and WIDOCO documentation links here as footnotes. https://github.com/STIInnsbruck/OntoCookie
Missing reference to Bollinger et al. in section 3.3 and if "et al" then use identity in plural form.
LLM should be spelt out when 1st mentioned.
Missing reference to Florea and Esteves.
Missing reference to Charlie (better have footnote than URL link).
Missing explanation and reference to UX antipatterns in section 4.2.
Missing reference to Rei in section 4.5.
Vector database == Graph Database? I would suggest using terminology more common for the semantic web community.
Missing references to ICO and EDPB in section 8.
The references need to be checked for proper and consistent capitalisation.

---

### Decision · Program_Chairs · 2024-08-02

Accept